# Mosquito midgut stem cell cellular defense response limits *Plasmodium* parasite infection

Ana-Beatriz F. Barletta [1] ✉, Jamie C. Smith [1], Emily Burkart[1], Simon Bondarenko[2], Igor V. Sharakhov [2], Frank Criscione[1], David O'Brochta[3] & Carolina Barillas-Mury [1] ✉

A novel cellular response of midgut progenitors (stem cells and enteroblasts) to *Plasmodium berghei* infection was investigated in *Anopheles stephensi*. The presence of developing oocysts triggers proliferation of midgut progenitors that is modulated by the Jak/STAT pathway and is proportional to the number of oocysts on individual midguts. The percentage of parasites in direct contact with enteroblasts increases over time, as progenitors proliferate. Silencing components of key signaling pathways through RNA interference (RNAi) that enhance proliferation of progenitor cells significantly decreased oocyst numbers, while limiting proliferation of progenitors increased oocyst survival. Live imaging revealed that enteroblasts interact directly with oocysts and eliminate them. Midgut progenitors sense the presence of *Plasmodium* oocysts and mount a cellular defense response that involves extensive proliferation and tissue remodeling, followed by oocysts lysis and phagocytosis of parasite remnants by enteroblasts.

Mosquitoes become infected with *Plasmodium* parasites when they ingest blood from a malaria-infected host. Fertilization of female gametes takes place in the gut lumen, giving rise to motile ookinetes that must traverse the midgut and can be targeted by the mosquito complement-like defense response[1,2]. Those ookinetes that reach the midgut basal lamina transform into oocysts, a stage in which the parasite multiplies and grows continuously for 2–3 weeks. Mature oocysts rupture, releasing thousands of sporozoites that migrate to and invade the salivary gland, and are transmitted when the mosquito bites a new host.

Oocysts develop under the basal lamina of the mosquito midgut epithelium and their surface is covered by a capsule composed of proteins derived from mosquitoes and the parasite[3] that is thought to conceal them from the mosquito immune system. Indeed, there is clear evidence that the mosquito complement-like system does not target the oocyst stage[4]. The oocyst is thought to be a "quiet"

developmental stage in which the parasite multiples continuously within the capsule. The architecture of the midgut epithelium is intricate in dipteran insects, consisting of at least four distinct cell types: stem cells, which are pluripotent and exhibit high levels of delta protein[5]; enteroblasts, the partially differentiated cells, that no longer express delta protein but have yet to reach full differentiation[6]. Enteroblasts then give rise to fully differentiated enterocytes and enteroendocrine cells[6]. Enterocytes, the most abundant type, have microvilli and serve both digestive and absorptive functions, while enteroendocrine cells, are involved in hormone-secretion.

Midgut stem cells maintain tissue integrity and homeostasis by self-renewing through asymmetric division. In this process, one daughter cell retains stem cell properties while the other—known as an enteroblast—becomes committed to differentiation[7,8]. Upon injury, these intestinal stem cells undergo further asymmetric divisions until tissue homeostasis is restored. Most injuries are transient and resolve

[1]Laboratory of Malaria and Vector Research, National Institutes of Allergy and Infectious Diseases, National Institutes of Health, Rockville, MD 20852, USA. [2]Department of Entomology, Virginia Polytechnic Institute and State University, Blacksburg, VA 24060, USA. [3]Institute for Bioscience and Biotechnology Research and Department of Entomology University of Maryland-College Park, Rockville, MD 20850, USA. ✉e-mail: anabeatriz.barlettaferreira@nih.gov; cbarillas@niaid.nih.gov

themselves, leading to a return to the midgut's baseline state[7,8]. In *Drosophila*, the Jak/STAT pathway plays a pivotal role in controlling midgut stem cell behavior. Its activation triggers both the proliferation of existing stem cells and the differentiation of enteroblasts within the tissue[9].

Here, we show that midgut stem cells respond to the presence of developing oocysts and proliferate to surround them and actively eliminate them. This novel stem cell-mediated defense response is modulated by the Jak/STAT pathway and results in drastic changes in midgut architecture.

## Results and Discussion

An *Anopheles stephensi* SDA-500 transgenic line (HP10) that expresses a fluorescent reporter (tdTomato) in a subset of midgut cells with morphology reminiscent of *Drosophila* midgut stem cells[10] (Fig. 1A) and in hemocytes (Fig. S1), was selected during an enhancer-trap screen. The GAL4 insertion was mapped to chromosome 3 (position 22,556,999 bp, Fig S2A–C), and a single chromosomal insertion in Chr 3 (div. 41B) was confirmed by fluorescence in situ hybridization (FISH) (Fig S2B). The reporter protein is expressed in small triangular midgut cells in the basal plane (potential intestinal stem cells, ISCs) (Fig. 1A, front and lateral view) and in a subset of larger cells embedded in the epithelium (potential committed enteroblasts) (Fig. 1B, front and lateral view) that lack microvilli and often do not reach the surface of the gut lumen. In *Drosophila*, ISCs divide asymmetrically to produce an ISC and an enteroblasts that exits the cell cycle and differentiate into enterocytes or enteroendocrine cells[11]. To define the identity of tdTomato-expressing cells, mosquito midguts were stained for delta, a classic marker of pluripotency in ISC[12]. The small triangular basal cells were positive for delta while cells embedded in the epithelia did not express delta, reflecting commitment to enteroblast differentiation (Fig. 1C, D). As expected, oral administration of bleomycin, a chemical that causes tissue damage[13], triggered an increase in tdTomato positive cells (Fig. S3A, B), indicative of proliferation of midgut progenitors to restore damaged epithelial cells. Furthermore, a fraction of dissociated midgut cells enriched for tdTomato-positive cells (Fig. S3C), with tdTomato mRNA levels 15-fold higher than whole midguts, was also enriched for delta and klumpfuss (*klu*) expression, markers of midgut stem cells and enteroblasts, respectively (Fig. S3C). In contrast, expression of the *An. stephensi* ortholog of the *Drosophila* POU box gene (*Pdm-1*), a marker of mature enterocytes[14], was not enriched (Fig S3C). The typical morphology, the response to chemical damage and the enrichment of gene markers, all indicate that the tdTomato-positive midgut cells in this transgenic line are epithelial progenitor cells (stem cells and enteroblasts).

Ookinete traversal causes irreversible damage to invaded mosquito midgut cells, which undergo apoptosis and are extruded into the midgut lumen[15]. *P. berghei* infection resulted in no statistically significant difference in the number of midgut progenitors one day post-infection (PI) compared to uninfected midguts from blood-fed females (Fig. 1E, H, Fig S4A), a time when ookinetes are traversing the midgut, and very few ookinetes (4%) were associated with progenitor cells (Fig. 1K). At 5 days PI there was a 3.5-fold increase in the number of progenitors in infected midguts (Fig. 1F, I, Fig. S4B; Mann-Whitney, *p* < 0.0001) and 46% of oocysts were in contact with at least one progenitor (Fig. 1L). Intense proliferation and differentiation of midgut progenitors occurred between days 5 and 10 PI (Fig. 1G, J), resulting in a 6.74-fold increase in the number of progenitors in infected midguts compared to blood-fed uninfected midguts (Fig. 1J and Fig. S4C and S5; Mann-Whitney, *p* < 0.0001). At 10 days PI most *Plasmodium* oocysts (85%) were in direct contact with midgut progenitors (Fig. 1G, M). The *Plasmodium*-induced increase in the number of enteroblasts and the displacement of epithelial cells by growing oocysts result in a drastic reorganization of the midgut epithelium (Fig. 2A, B). Progenitor cells proliferate forming "ribbons" of enteroblasts intercalated between

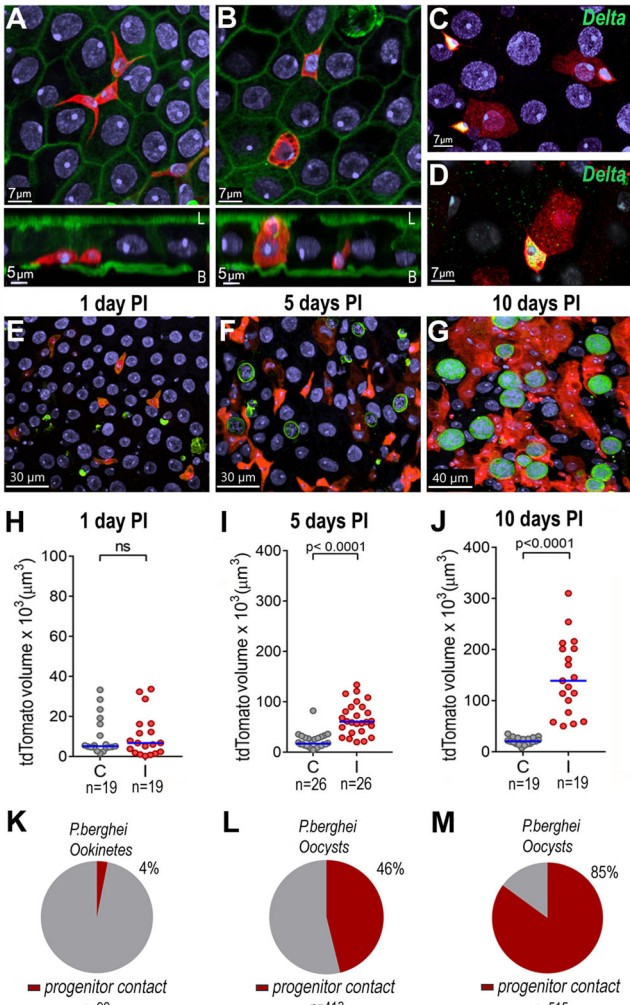

**Fig. 1 | Response of *Anopheles stephensi* HP10 line midgut progenitors to *Plasmodium berghei* infection. A** Small triangular tdTomato⁺ cell in the basal side of the midgut epithelium. **B** Larger tdTomato⁺ cells embedded in the epithelium. Front and lateral views. Scale bar: 7 and 5 μm, respectively. td-Tomato (red), nuclei (blue) and actin (green). L = lumen and B = basal. **C** Delta antibody staining of small triangular tdTomato⁺ cells. **D** Close-up of delta immunostaining in small triangular cells. tdTomato in red; Nuclei in blue and delta in green. Scale bars: 7 μm. Micrographs are representative of experiments that were independently reproduced at least 2 times with multiple individuals each experimental group obtained similar results. tdTomato⁺ cells in midguts (**E**) 24 h, (**F**) 5 days and (**G**) 10 days PI. Parasites (green), tdTomato (red) and nuclei (blue). Volume of tdTomato (tdTomato⁺ cells) in the mosquito midgut at (**H**) 24 h (*p* = 0.9770) (**I**) 5 days (*p* < 0.0001) and (**J**) 10 days PI (*p* < 0.0001). Each dot represents the volume of red fluorescence for individual midguts and the medians are indicated with the horizontal line. Two-tailed Mann Whitney U test, ns = *p* > 0.05. Percentage of *Plasmodium* parasites that are in contact with midgut progenitors at (**K**) 24 h, (**L**) 5 days and (**M**) 10 days PI. C = uninfected midguts, I = infected midguts. n = number of parasites. The numerical data underlying the plots in the manuscript are provided as a Source Data File.

mature epithelial cells (Fig. 2B, C) that often surround and come in direct contact with developing oocysts (Fig. 2D, F).

We explored whether cell damage by ookinete invasion was sufficient to elicit proliferation of midgut progenitors, or whether the response required the presence of developing oocysts. Two groups of mosquitoes were fed on the same infected mouse and kept at a permissive temperature (21 °C) for *P. berghei* for 48 h, to allow ookinete development and midgut invasion. One group was then shifted to a non-permissive temperature (28 °C) 2 days PI to disrupt further oocyst development, while oocysts were allowed to mature at 21 °C in the

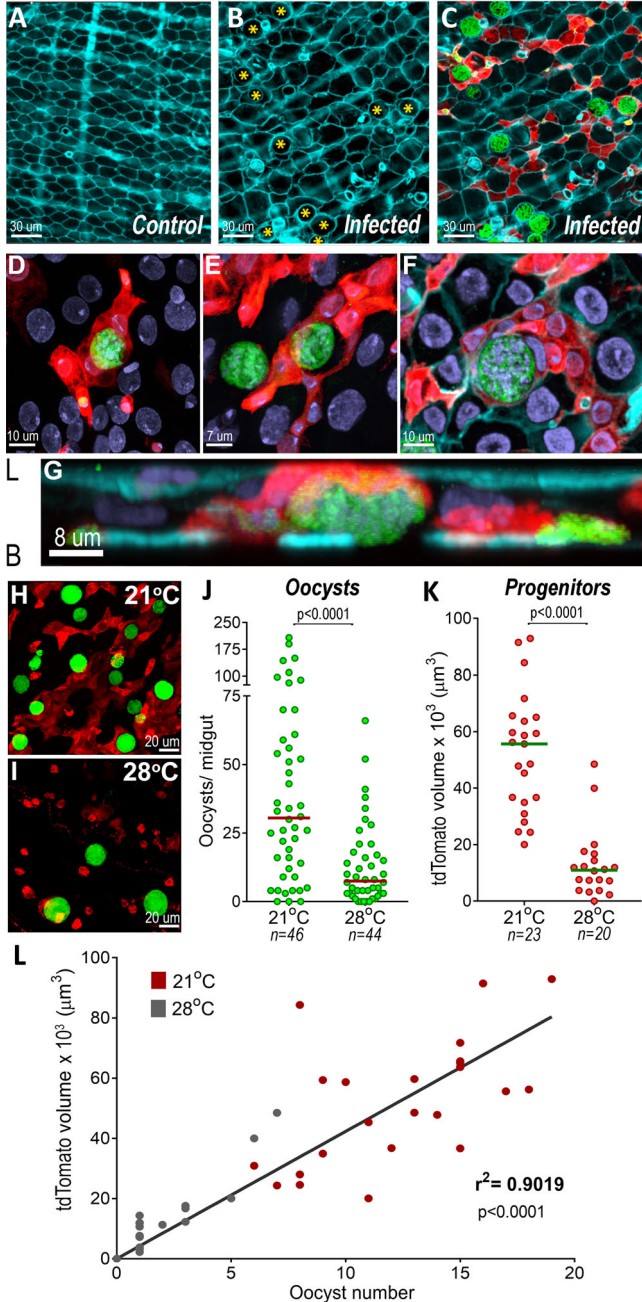

**Fig. 2 | Effect of *Plasmodium* infection on midgut architecture and of oocysts intensity of infection on proliferation of midgut progenitors. A** Actin staining of blood-fed control midguts. **B** Actin staining in *P. berghei*-infected midguts 10 days PI. Oocysts in the epithelia are indicated by yellow asterisks. **C** Actin staining (cyan), progenitors (red) and *P.berghei* oocysts (green). Scale bar: 30 μm. **D, E** Progenitor "ribbons" (red) surrounding oocysts (green), 10 days PI. **F** Actin (cyan) staining showing close association of progenitors (red) embedded in the epithelium with oocysts (green), 10 days PI. Scale bar: 10 μm. **G** Lateral view of (**F**), showing midgut progenitors (red) surrounding an oocyst 10 days PI. L = lumen and B = Basal. Micrographs are representative of experiments that were independently reproduced at least 2 times with multiple individuals in each experimental group obtained similar results. **H** Midgut progenitors and oocysts in the midgut of *An. stephensi* kept at (**H**) 21 °C and (**I**) 28 °C. **J** Oocysts counts of mosquitoes kept at 21 °C, a permissive temperature for oocyst development, and at 28 °C, a non-permissive one. Each dot represents the number of oocysts on individual midguts. The median is indicated by the horizontal line. Two-tailed Mann-Whitney U test ($p < 0.0001$). **K** Volume of tdTomato (tdTomato⁺ cells) 10 days PI of females kept at 21 °C and 28 °C. Each dot represents the volume of red fluorescence for individual midguts and the medians are indicated by the horizontal line. Two-tailed Mann Whitney U test ($p < 0.0001$). **L** Correlation of oocyst numbers and tdTomato volume (tdTomato⁺ cells) of midguts from females kept at 21 °C (red dots) and 28 °C (gray dots). Each dot represents an individual mosquito. Linear regression, $r^2 = 0.9019$, $p < 0.0001$. The numerical data underlying the plots in the manuscript are provided as a Source Data File.

stabilizes *TEP1*, in *An. stephensi* SDA-500 does not significantly affect *P. berghei* survival[16]. We confirmed that silencing *TEP1* in the *An. stephensi* HP10 transgenic line also had no statistically significant effect on parasite survival ($p = 0.5802$, Fig. S7A), indicating that *TEP1*-mediated mosquito complement immunity is not limiting *P. berghei* survival. However, we documented a statistically significant decrease between the number of oocysts present at 2 and 8 days PI (Mann-Whitney, $p < 0.0001$, Fig. S7B), suggesting that a substantial number of oocysts may be actively eliminated by mosquito defenses, as previously observed in *An. gambiae*[17].

Activation of JAK-STAT signaling in ISCs promotes rapid division and differentiation of progenitors[9]. Overactivation of JAK/STAT signaling in *An. stephensi* HP10 females by silencing *SOCS* (Suppressor of Cytokine Signaling), an inhibitor of the JAK-STAT pathway, resulted in a statistically significantly increase in the number of midgut progenitors 10 days post-infection (PI) (Mann-Whitney, $p < 0.0013$, Fig. 3A–C). A concomitant reduction in the number of oocysts (Mann-Whitney, $p < 0.0004$, Fig. 3D) relative to dslacZ controls was observed, similar to what has been reported in *An. gambiae*[17]. Conversely, disrupting JAK/STAT signaling by silencing the JAK kinase Hopscotch (*HOP*) greatly reduced proliferation of midgut progenitors (Mann-Whitney, $p < 0.0001$, Fig. 3E–G) and significantly increased the number of oocysts (Mann-Whitney, $p < 0.009$, Fig. 3H). Furthermore, reducing delta expression promoted proliferation and differentiation of midgut progenitors (Mann-Whitney, $p < 0.0004$, Fig. 3I–J), and resulted in a statistically significant decrease in oocyst numbers (Mann-Whitney, $p < 0.0003$, Fig. 3L). Altogether, we conclude from these data that proliferation of midgut progenitors is detrimental to oocyst survival.

The hypothesis that midgut progenitors interact directly with *Plasmodium* oocysts to eliminate them was explored. Live confocal imaging was used to directly image midgut progenitor cells of live infected mosquitoes 10 and 14 days PI for periods of 9–12 h. In females imaged 10 days PI, midgut progenitors were observed extending pseudopodia-like extensions that came in direct contact with the parasite's surface and exerted pressure on the oocyst surface until the cell integrity was compromised and the GFP label in the oocyst cytoplasm was released (Fig. 4A and Supplementary Movies 1 and 2). At 14 days PI, we observed a larger oocyst being pressed on either side by two progenitor cells that also formed pseudopodia-like extensions, deforming the parasite and releasing the fluorescent cell cytoplasm towards the gut lumen (Fig. 4B and Supplementary Movies 3, 4). Dead

second group. As expected, the number of oocysts 10 days PI decrease significantly at a higher temperature (Mann-Whitney, $p < 0.0001$, Fig. 2H–J). Although similar ookinete invasion took place in both groups, decreasing oocyst numbers significantly reduced proliferation of midgut progenitors (Fig. 2K). Furthermore, there was a strong correlation ($r^2 = 0.902$, $p < 0.0001$) between the volume of midgut progenitors on individual midguts and the number of oocysts, even at low oocyst density (1–20 oocysts/midgut) (Fig. 2L), suggesting that the observed proliferation of midgut progenitors is a response to the presence of oocysts. A strong positive correlation between volume of midgut progenitors and oocysts numbers was also observed when mosquitoes from two different experiments were combined and those kept at 21 °C ($r^2 = 0.8018$, $p < 0.0001$) were analyzed separately from those kept ant 28 °C ($r^2 = 0.8534$, $p < 0.0001$) (Fig. S6).

The thioester-containing protein 1 (*TEP1*) is a key effector of complement-mediated elimination of *P. berghei* ookinetes in *An. gambiae*[1]. However, silencing *LRIM1*, a gene encoding a protein that

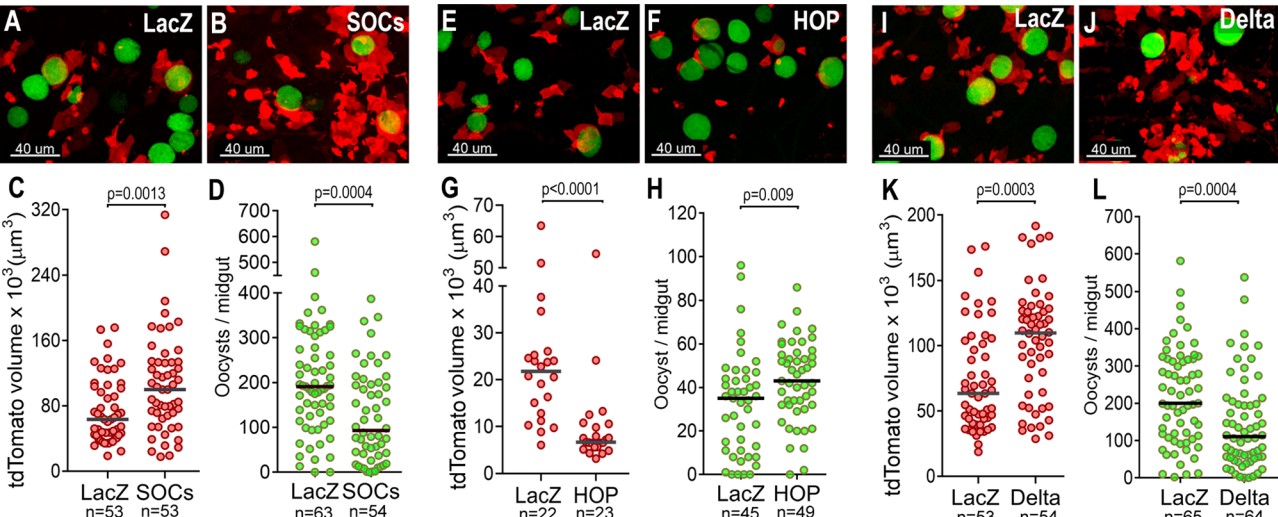

**Fig. 3 | Effect of proliferation of midgut progenitors on oocysts survival. A** LacZ and (**B**) *SOCS*-silenced midguts 10 days PI. **C** Volume of tdTomato (tdTomato+ cells) in lacZ and *SOCS*-silenced midguts 10 days PI ($p = 0.0013$). **D** Oocyst counts of lacZ, and *SOCS* silenced midguts 10 days PI ($p = 0.0004$). **E** lacZ and (**F**) *HOP*-silenced midguts 10 days PI. **G** Volume of tdTomato (tdTomato+ cells) in lacZ and *HOP*-silenced midguts 10 days PI ($p < 0.0001$). **H** Oocyst counts of lacZ and *HOP*-silenced midguts 10 days PI. **I** LacZ and (**J**) delta-silenced midguts 10 days PI ($p = 0.009$). **K** Volume of tdTomato (tdTomato+ cells) in lacZ and delta-silenced midguts

10 days PI ($p = 0.0003$). **L** Oocyst counts of lacZ and delta-silenced midguts 10 days PI ($p = 0.0004$). Each dot represents an individual mosquito. Median is indicated by the horizontal line. Two-tailed Mann Whitney U test. Midgut progenitors are shown in red and *P. berghei* oocysts in green. Scale bars: 40 μm. The numerical data underlying the plots in the manuscript are provided as a Source Data File. Micrographs are representative of experiments that were independently reproduced at least 2 times with multiple individuals in each experimental group and obtained similar results.

oocysts and fragments positive for GFP in immunofluorescence staining were often observed 5- and 10-days PI inside tdTomato⁺ cells embedded in the epithelium (Fig. 4C–E and Fig. S8A–C), showing that midgut enteroblasts internalize dead parasites. Furthermore, large live oocysts exhibit surface staining with the LIVE/DEAD® dye, while dead oocysts are smaller, have strong signal inside the capsule and are also frequently found within midgut enteroblasts (Fig. 4F–H). At day 5 PI, 69% of oocysts in contact with midgut progenitors are dead (Table S1).

*Plasmodium* infection triggered intense proliferation of midgut progenitors, which allowed enteroblasts to come in direct contact with developing oocysts. Promoting proliferation of progenitor cells by over activating STAT signaling or by reducing delta expression, reduced oocyst survival. We have previously described a late-phase response mediated by the JAK/STAT pathway in *An. gambiae* with an unknown effector mechanism[17]. Here, we provide direct evidence that, besides their critical role in maintaining the integrity of the midgut epithelial barrier in response to cell damage, stem cell-derived *An. stephensi* enteroblasts are effectors of a JAK/STAT-mediated cellular defense response against *Plasmodium* oocysts.

It is not clear how midgut progenitors detect the presence of oocysts, as initially (24 h PF) they do not colocalize. One can envision that oocysts may secrete glycosyl-phosphatidyl-inositol (GPI) that is detected directly by progenitor cells. Alternatively, GPI may activate enterocytes near the parasite and they, in turn, may release a secondary signal(s) detected by midgut progenitors. As oocysts increase in size, they may also exert physical pressure on adjacent epithelial cells and trigger the release of chemokines that promote proliferation of midgut progenitors. Live imaging indicates that midgut progenitors can move, albeit slowly, and cooperate to compress oocysts and extend pseudopod-like projections that come in direct contact with the oocysts surface before their integrity is compromised, suggesting that they release a local factor(s) that damages the oocyst capsule. It also remains to be established whether these are universal responses of midgut progenitors that are also triggered by infection with human *Plasmodium* parasites. Stem cell migration plays a critical role in the effective regeneration of the adult *Drosophila* midgut.

Enteroendocrine cells guide the direction of stem cell movement, while enteroblasts support migration by triggering signaling pathways[18]. Oocysts fragments are often found inside midgut enteroblasts, indicating that they internalize dead parasites. At early stages of *Drosophila* embryonic development, dying cells are mostly engulfed by sister cells that are not fully differentiated[19], indicating that epithelial cells with some degree of pluripotency can act as phagocytes. Taken together, our studies provide direct evidence that midgut progenitors can detect the presence of *Plasmodium* oocysts and mount a cellular defense response that involves extensive proliferation and tissue remodeling, followed by oocyst lysis and phagocytosis of parasite remnants.

## Methods

### Ethics statement
Public Health Service Animal Welfare Assurance #A4149-01 guidelines were followed according to the National Institutes of Health Animal (NIH) Office of Animal Care and Use (OACU). These studies were done according to the NIH animal study protocol (ASP) approved by the NIH Animal Care and User Committee (ACUC), with approval ID ASP-LMVR5.

### *Anopheles stephensi* HP10 hindgut transgenic line
HP10 line was generated using an enhancer trap system described by ref. 20. Briefly, *An. stephensi* SDA-500 embryos were injected with a *piggyBac* transposon-based promoter-less GAL4 enhancer trap element that is remobilized by a *piggyBac* transposase expressed in trans to the gene cassette. These results in random remobilization of the GAL4-containing element. We established remobilized enhancer trap lines that displayed GAL4 expression, specifically in adult hemocytes and midgut stem cells. Tissue-specific expression of GAL4 can be visualized by indirect immunofluorescence using anti-GAL4 antibodies or by crossing HP10 to lines containing reporter genes under the regulatory control of GAL4 responsive promoters. In this manuscript, we used a crossing of the HP10 line to another mosquito line that had a reporter gene (UAS-Td-tomato) under the regulatory control of

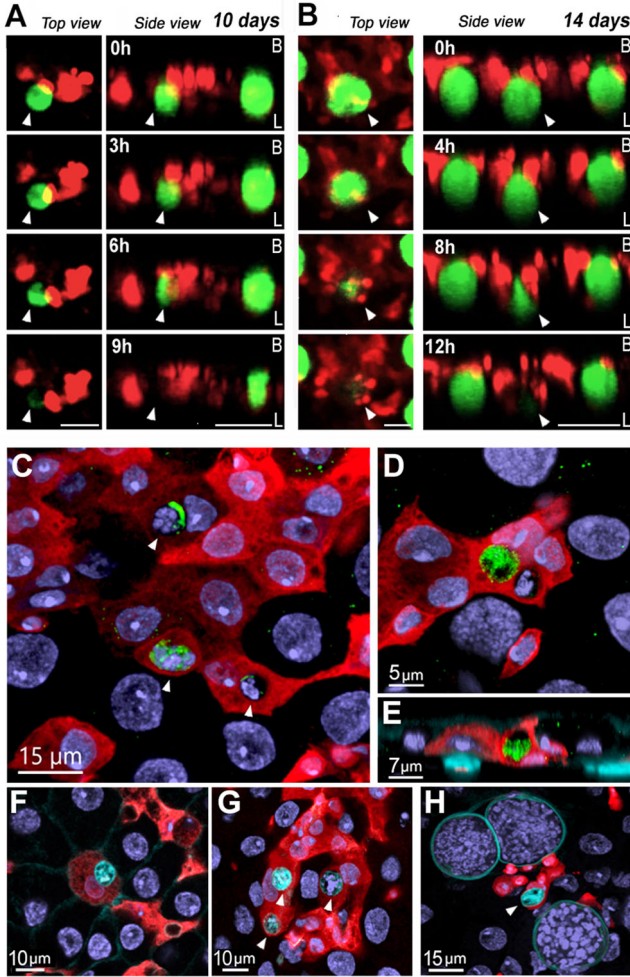

**Fig. 4 | Midgut progenitors actively eliminates *Plasmodium* oocysts and internalize parasite remnants. A** Whole mosquito live imaging over the course of 9 h 10 days pi. Scale bar from top view: 20 μm and from side view: 30 μm. **B** Whole mosquito live imaging over the course of 12 h 14 days PI. Scale bar from top view: 20 μm and from side view: 50 μm. **C** Oocyst fragments inside midgut progenitors at 5 days PI. White arrows indicate parasite remnants inside of midgut progenitors. Scale Bar: 15 μm. Leftover of *P.berghei* GFP from inside of a midgut progenitor 5 days PI (**D**) Front view and (**E**) Lateral view. Scale Bar: 5 μm and 7 μm respectively. Midgut progenitors are in red and *P.berghei* are in green (**F**) Dead oocyst in cyan inside of a midgut progenitor 5 days PI. Scale Bar: 10 μm. **G** Dead oocysts in cyan inside midgut progenitors 5 days PI. Scale Bar: 10 μm. **H** Live oocysts with cyan staining on the surface and a smaller dead oocyst with staining inside the capsule, engulfed by a midgut progenitor. White arrows indicate dead oocysts. Scale Bar: 15 μm. Micrographs are representative of experiments that were independently reproduced at least 2 times with multiple individuals in each experimental group and obtained similar results.

GAL4[20]. HP10 is maintained as homozygotes at 28 °C, 80% humidity under a 12 h light/ dark cycle, and kept with 10% Karo syrup solution during adult stages.

## Mouse feeding and *Plasmodium berghei* infection

Mosquito infections with *Plasmodium berghei* were performed using transgenic *An. stephensi* HP10 female mosquitoes and a transgenic GFP *P.berghei* parasite strain (ANKA GFPcon 259cl2) maintained by serial passages into 3 to 4-week-old female BALB/c mice (Charles River, Wilmington, MA, USA) from frozen stock. Mice were kept in individual ventilated cages at 20–21 °C and 56% humidity and under a strict 12 h light-dark cycle with sterile water and food provided *ad libitum*. Mouse

parasitemia was measured to determine the infectivity before feeding the mosquitoes. Four to five-day-old females were fed when mice reached 3–5% parasitemia for regular infections or before it reached 1% parasitemia for lower ones (oocyst counting experiments). Same-age uninfected mice were used to feed blood-fed control mosquitoes. After feeding, both control and infected mosquitoes were maintained at 19 °C, 80% humidity, and 12 h light/dark cycle until the day of dissection.

## Midgut Immunostaining

To image the midgut at late stages of infection, we fed mosquitoes a saline solution supplemented with 10% BSA (Bovine Serum Albumin) right before dissection to distend the midgut epithelia. A day before the dissection, a cup with water was placed inside the cage for egg laying. The next day, mosquitoes were artificially fed with 10% BSA in 0.15 M Sodium Chloride mixed with 10 mM Sodium Bicarbonate, pH 7.2[21]. Sodium Bicarbonate solution must be fresh, and pH adjusted on the day of the feeding.

Mosquitoes were dissected between 30 min and 1 h after feeding. For one day, PI timepoint midguts were already distended, and mosquitoes were not fed with saline solution. After feeding, midguts were dissected in PBS at room temperature and fixed for 30 s with 4% Paraformaldehyde (PFA) to preserve the midgut structure. Tissues were then opened longitudinally in ice-cold PBS and cleaned to remove the bolus. Then, tissues were placed in 4% PFA for at least one hour at room temperature. Tissues were washed with 0.1% Triton PBS three times, 10 min each, at room temperature. Midguts were blocked for 4 h in 2% BSA, 0.1% gelatin and 0.1% Triton in PBS at room temperature and stained overnight using either rabbit polyclonal anti-DsRed (1:1000) (Living Colors® DsRed Polyclonal Antibody, 632496, Takara Bio, San Jose, CA, USA) or mouse monoclonal anti-RFP (1:1000) (Rocklands Immunochemicals, Mouse Monoclonal 8E5.G7 IgG2a kappa, 200-301-379, Philadelphia, PA, USA) to visualize cells that expressed d-Tomato depending on antibody combination. To visualize ookinetes (1 day PI) in the midgut, we stained midguts using mouse anti-Pbs21 and to stain early oocysts (5 days PI) we used rabbit polyclonal anti-PbCap380 (1:1000). Anti-Pbs21 and PbCap380 were kindly provided by Dr. Marcelo Jacobs Lorena from Johns Hopkins Malaria Research Initiative (JHMRI). To visualize oocysts at later stages (10 days PI) we used its inner GFP fluorescence and to detect oocysts fragments inside the enteroblasts we incubated tissues with rabbit anti-GFP (1:500) (Anti-GFP antibody, ab6556, Abcam, Cambridge, UK). For Delta staining, midguts were incubated with mouse monoclonal anti-Delta (1:20) (Dl antibody, C594.9B, DSHB, Iowa City, IA, USA) in blocking solution (PBS containing 2% BSA, 0.1% gelatin and 0.1% triton). All primary antibody incubations were performed overnight at 4 °C.

After the primary antibody, midguts were washed with PBS containing 2% BSA, 0.1% gelatin, and 0.1% triton and incubated with secondary antibodies goat anti-rabbit (1:1000) (Alexa Fluor 488 (A-11008) or 594 (A-11012), Molecular Probes, ThermoFisher Scientific, Waltham, MA, USA) or goat anti-mouse (1:1000) (Alexa Fluor 488 (A-11001) or 594 (A-11005), Molecular Probes, ThermoFisher Scientific, Waltham, MA, USA) for 2 h at room temperature. Midguts were washed three times with blocking buffer and twice with PBS with 0.1% Triton. Tissues were incubated with 20 μM Hoechst 33342 (405, Molecular Probes, 62249, ThermoFisher Scientific, Waltham, MA, USA) for nuclei staining and 1 U of phalloidin (Alexa Fluor 647 (A22287) or Alexa Fluor 750 (A30105), Molecular Probes, ThermoFisher Scientific, Waltham, MA, USA) to visualize the midgut actin, diluted in PBS with 0.1% triton for 30 min at room temperature. Microscope slides were mounted using a drop of Prolong Gold Antifade Mountant (Molecular Probes, ThermoFisher Scientific, Waltham, MA, USA).

### Td-tomato volume quantification and surface analysis

To determine the volume of the cells expressing td-Tomato within the midgut we used the surface mode on Imaris 9.9.1 (Bitplane, Concord, MA, USA). To create a surface, we used confocal z-stack sections. First, we applied the Gaussian filter to smooth the picture, then we used the surface mode with a threshold of 16 for the number of voxels. After the surfaces were generated, we applied a refining filter to remove smaller volumes and eventual tracheal signal. From the surface created from the td-Tomato signal we were able to quantify the volume of the midgut progenitors in the tissue. To determine the association between midgut progenitors and the parasite at different stages of infection. We also created a surface for the parasite fluorescent signal. We smooth the picture as described before and then created a surface with a threshold of 45 for number of voxels. Small surfaces were eliminated with a second filter to keep only the surfaces correspondent to parasites. We counted the number of parasite surfaces and how many were associated with the surfaces of midgut progenitors.

### Midgut oocyst counting

*Plasmodium berghei* infections were evaluated by counting oocyst numbers per mosquito midgut after feeding on an infected mouse. Infected mosquitoes were kept at 19 °C for ten days after feeding when dissected, and their midgut was fixed in 4% PFA for 15 min at room temperature. After washing with PBS three times, midguts were mounted in a slide and counted under a fluorescence microscope, where live oocysts were identified by their GFP expression. The number of oocysts per midgut was represented in a scatter plot where each dot represents an individual mosquito. For the experiment to count oocysts 2 and 8 days after infection (Fig. S5B), we used a *P.berghei* mCherry strain, which is brighter than GFP and more straightforward to measure in the early days of mosquito infection without the need for antibody staining. Two days post-infection fed mosquitoes kept at 19 °C still had a blood bolus. Therefore, they were opened and cleaned before fixation with 4% PFA, as described above. For oocyst counting, midguts were mounted using VECTASHIELD® Antifade Mounting Medium with DAPI (H-1200-10, Vector Laboratories, Newark, CA, USA).

### Confocal microscopy

Confocal images were captured using a Leica TCS SP8 (DM8000) confocal microscope (Leica Microsystems, Wetzlar, Germany) with either a 40 × or a 63 × oil immersion objective equipped with a photomultiplier tube/ hybrid detector. Midguts were visualized with a white light laser, using 498 nm excitation for Alexa 488 (phalloidin or *P.berghei* parasites), 588 nm excitation for Alexa 594 (midgut progenitors), 644 nm excitation for Alexa 647 (Live/Dead probe and phalloidin) and a 405 nm diode laser for nuclei staining (Hoechst 33342). Images were taken using sequential mode and variable z-steps. Image processing and merging were performed using Imaris 9.9.1 (Bitplane, Concord, MA, USA) and Adobe Photoshop CC (Adobe Systems, San Jose, CA, USA).

### Mapping GAL4 enhancer-trap insertion

Splinkerette PCR was used to map the genomic location of the GAL4 enhancer-trap insertion[22]. Briefly, genomic DNA was isolated from individual *An. stephensi* HP10 larvae (4th instar). The genomic DNA was digested by BstYI to produce sticky ends. A double stranded splinkerette oligonucleotide with stable hairpin loop and compatible sticky ends is ligated to the digested genomic DNA (SPLNK-GATC-TOP GATCCCACTAGTGTCGACACCAGTCTCTAATTTTTTTTTTCAAAAAAA and SPLNK-BOT−CGAAGAGTAACCGTTGCTAGGAGAGACCGTGGCTG AATGAGACTGGTGTCGACACTAGTGG). Next, we perform two rounds of nested PCR. For the first round, we used the following primers: SPLNK#1- CGAAGAGTAACCGTTGCTAGGAGAGACC and 3'SPLNK-PB#1−GTTTGTTGAATTTATTATTAGTATGTAAG (to map three prime

end) or 5'SPLNK-PB#1- ACCGCATTGACAAGCACG (to map five prime end). For the second round, we used the following primers: SPLNK#2− GTGGCTGAATGAGACTGGTGTCGAC and 3'SPLNK#2−GGATGTCTCT TGCCGAC (to map three prime ends) or 5'SPLNK-PB#2−CTCCAAG CGGCGACTGAG (to map five prime ends). Those two rounds generate a PCR fragment that contains the flanking genomic DNA between the *piggybac* element insertion site and the genomic digestion site. Detailed protocol is described in ref. [22]. PCR fragments generated from the three and the five prime ends were cloned in TOPO-TA vector (K450002, ThermoFisher Scientific, Waltham, MA, USA) and then used for a standard Sanger sequencing reaction using TOPO vector primers.

### Fluorescence in situ hybridization of the GAL4-specific fluorescent probe with the polytene chromosomes of *An. stephensi* HP10 transgenic line

Fluorescence in situ hybridization (FISH) was done following the previously published protocol with minor modifications[23].

### Fluorescent DNA probe preparation

Genomic DNA was extracted from *An. stephensi* HP10 hindgut mosquitoes and amplified with GAL4-specific primers (Gal4clone_F− AAGAAAAACCGAAGTGCGCC and Gal4clone_R−CACCAAACAAAGCA-GACGGG). About 30 ng of the purified PCR product was used in a Random-primer labeling reaction, in which Cyanine 5-dUTP (Enzo Life Sciences Inc., Ann Arbor, MI, USA) was incorporated into the DNA by Klenow fragment (ThermoScientific, Graiciuno, Lithuania) and Random Primers DNA Labeling System (Invitrogen, Carlsbad, CA, USA). The manufacture-supplied protocol for the Random Primers DNA Labeling System was used. The labeled DNA was precipitated by 2.5 volumes of 96% Ethanol and 0.1 volume of 3 M sodium acetate at −20 °C overnight and then centrifuged at 14,000 g, 4 °C for 20 min. The supernatant was removed, and the pellet of DNA was air-dried and dissolved in 50 µl of hybridization buffer (60% deionized formamide, 2 × SSC, 10% dextran sulfate) by shaking the mix in the Eppendorf ThermoMixer C (MilliporeSigma, St. Louis, MO, USA) at 2000 rpm, 37 °C for 1 h.

### Polytene chromosome preparation

*Anopheles stephensi* HP10 females were fed with blood twice after emergence and allowed to lay eggs. A third blood meal was offered, and females were allowed to develop their ovaries for 26 h at 26 °C and 80% humidity. Ovaries were dissected from half-gravid females under a MZ6 Leica stereo microscope (Leica Camera, Wetzlar, Germany) and fixed in fresh modified Carnoy's solution: methanol: glacial acetic acid (3:1). The ovaries were kept at room temperature overnight until they were transferred ovaries to −20 °C for a long-term storage. Parts of dissected ovaries were put in drops of 50% propionic acid on slides for about 5 min until follicles became clear and about twice their original size. Follicles were separated from each other with needles, and other tissues were removed by wiping them away with a paper towel. A fresh drop of 50% propionic acid was applied to the separated follicles. Follicles were covered with a dust-free coverslip and left for about 5 min. A piece of filter paper was placed over the coverslip that was gently tapped with a pencil eraser to release polytene chromosomes from the nurse cells of follicles. The banding pattern and spreading of polytene chromosomes were examined using an Olympus CX43 Phase Microscope (Olympus, Tokyo, Japan) with a 20 × objective. Slides with suitable chromosomal preparations were placed in a humid chamber with 4 × SSC in the bottom of the chamber and incubated at 4 °C overnight for better flattening of chromosomes. Slides were immersed in liquid nitrogen until the bubbling stopped (10–15 s). After taking the slides out of the liquid nitrogen, the coverslips were removed with a razor blade. Slides were immediately placed in a slide jar with pre-chilled 50% ethanol (−20 °C) and kept at 4 °C for at least 2 h. The preparations in a slide jar were dehydrated with ethanol series of 70%

and 90% for 5 min each at 4 °C and then with 100% ethanol for 5 min at room temperature. Slides were air-dried and kept in a box until the hybridization.

## Fluorescence in situ hybridization

Chromosomal preparations were incubated in the 2× SSC with 4% formaldehyde solution at 60 °C for 30 min. Then, 15 µl of the labeled GAL4-specific fluorescent DNA-probe dissolved in the hybridization buffer were applied to the slide and covered with a 22 × 22 coverslip. The borders of the coverslip were insulated with rubber cement. Slides were then placed into the Thermobrite machine (Leica Biosystems, Wetzlar, Germany) and heated at 80 °C for 5 min to denature chromosomal DNA and the probe. Hybridization occurred at 37 °C overnight. On the next day, the slides were washed in 2 × SSC at 60 °C for 15 min, then in 2 × SSC at room temperature for 15 min, and finally in 0.2 × SSC for 10 min. The washing solution was removed from the slide, and 15 µl of Prolong Gold Antifade Mounting with DAPI (ThermoFisher Scientific, Waltham, MA, USA) was applied to the slide and covered with a 22 × 22 coverslip. Microscopy and image acquisition were performed with an Axio Imager Z1 microscope (Carl Zeiss MicroImaging GmbH, Munich, Germany). Image processing was performed by the Fiji software (cite: doi:10.1038/nmeth.2019). Mapping of the location of the probe was done according to the standard cytogenetic map of polytene chromosomes for *An. stephensi*[24].

## Midgut chemical damage by Bleomycin treatment

To induce chemical damage in the midgut, 3- to 4 day-old females of *An. stephensi* HP10 hindgut line were fed with 10% Karo syrup solution supplemented with 25 µg/ml Bleomycin (Bleomycin sulfate, anticancer and antibiotic agent, ab142977, Abcam, Cambridge, UK) for two days[25]. The fresh solution was offered every day of the treatment. After two days, females were artificially fed with 10% BSA saline solution to distend the midgut tissue as described above.

## Statistical analysis and reproducibility

All statistical analyses were performed using GraphPad Prism 5 (GraphPad, San Diego, CA, USA). Statistical analysis, where pertinent, was conducted using either the student Unpaired t-test, Mann-Whitney Test, ANOVA Dunnett multiple comparisons, and ANOVA Kruskal-Wallis test and are indicated in the legend of each figure when appropriate. The test choice was determined based on the variation of the sample, whether parametric or non-parametric, and the number of treatments in each experiment. Significance was assessed at $p < 0.05$. The error bars represent the Standard Error of the Mean (SEM). Each experiment was performed independently at least 2 times with 3 biological replicates for each experimental group. Microscopy experiments were also independently performed at least 2 times with 10 individuals for each group. Independent replicates reproduced similar results.

## Enrichment of midgut progenitors for RNA extraction

*Anopheles stephensi* HP10 hindgut females were dissected, and 15 midguts were transferred to a tube containing 100 µl of elastase solution in PBS (1 mg/ml) (07453, Stem Cell Technologies, Vancouver, British Columbia, Canada) to dissociate the midgut tissue. Incubate the tubes in a shaker for 1 h at 27 °C, pausing to pipette up and down every 15 min with a siliconized pipette tip to avoid tissues sticking. After incubation, 100 µl of 2% BSA solution in PBS was added to the mix to stop digestion. Digested tissue was then loaded into a 20 µm cell strainer (pluriStrainer Mini 20 µm, pluriSelect, Leipzig, Germany). We passed the filtrate two more times on the same filter with a new pipette tip every time. We added 900 µl of TRIzol LS (TRIzol™ LS Reagent, 10296010, ThermoFisher Scientific, Waltham, MA, USA) and proceeded to RNA extraction. We compared the RNA expression of midgut enriched fractions with RNA from whole midguts placed straight into 1 ml of TRIzol.

## RNA extraction, cDNA synthesis, and qPCR analysis

Pools of 15 whole midguts from *An. stephensi* HP10 line were collected from sugar-fed females and placed directly into 1 ml TRIzol reagent (TRIzol™ reagent, 15596026, ThermoFisher Scientific, Waltham, MA, USA) and homogenized with a motorized pestle. The midgut-enriched fractions were placed in TRIzol LS as described above, and RNA extraction was conducted as follows. Two hundred microliters of chloroform were added (1/5 of TRIzol volume) to 1 mL of TRIzol and vortex vigorously for the aqueous phase separation. Samples were centrifuged for 12,000 RCF, 10 min, 4 °C. The aqueous phase was then transferred to a fresh 1.5 mL Eppendorf tube, and the RNA precipitated by adding 0.25 mL of isopropyl alcohol (500 mL per 1 mL TRIZOL reagent used). Five microliters of linear acrylamide (5 mg/mL) were added to each tube to aid in precipitation and pelleting. Samples were mixed by repeated inversion ten times, incubated for 10 min at room temperature, and spun at 12,000 RCF, 10 min, 4 °C. All the supernatant was removed, and the RNA pellets were washed twice with 75% ethanol (minimum 1 mL per 1 mL of TRIZOL). Tubes were mixed by vortexing and centrifuged at 7500 RCF, 5 min, 4 °C to wash the pellets. After the last wash, the supernatant was removed, and samples were air-dried until almost dry but not thoroughly (still translucent). RNA was solubilized with 30 µL of RNAse-free water, pipetting a few times to homogenize, and then placed at 55 °C for 10 min to resuspend the RNA. RNA concentration was measured at 260 nm using a Denovix Spectrophotometer (DS-11 Series Spectrophotometer/Fluorometer). One microgram of total RNA was used for complementary DNA (cDNA) synthesis using Quantitect Reverse Transcription Kit (Qiagen, Germantown, MD, USA) according to the manufacturer's instructions. Gene expression was assessed by quantitative PCR (qPCR) using the resulting cDNA as a template. Quantitative PCR (qPCR) was used to measure td-Tomato (AY678269.1), *delta* (ASTE009642 or ASTEI20_036824), *klumpfuss* (Klu) (ASTE009884 or ASTEI20_041957) and *POU domain transcription factor* (Pdm) (ASTE011391 or ASTEI20_034725) gene expression in whole sugar-fed midguts and fractions enriched for midgut progenitors. We used the DyNamo SYBR green qPCR kit (ThermoFisher Scientific, Waltham, MA, USA) with specific primers, and the assay ran on a CFX96 Real-Time PCR Detection System (Bio-Rad, Hercules, CA, USA). A 133-bp fragment was amplified for td-Tomato (F- ATCGTGGAACAGTACGAGCG and R- TGAACTCTTTGATGACGGCCA). A 165-bp fragment was amplified for *delta* (F- TGGGAGTTTCAACCGACTGG and R- CGATCGGTGAG-CAGGTGTAA). A 154 bp fragment was amplified for *Klu* (F- AGTCTC-CACAGCAACCGATG and R- CGGGCAAACTCCTGGTAGAG). A 134-bp fragment was amplified for *Pdm* (F- GCCTATCCTCACCTTCGTCC and R- CGGTCATTCCTGCTTGATGC). Relative expression was normalized against *An. stephensi* ribosomal protein S7 (RpS7) as internal standard and analyzed using the ΔΔ Ct method[26,27]. RpS7 (ASTE004816 or ASTEI20_033711) primers sequences were: F- TGGAAATGAACTCG-GATCTGAAG and R−CCTTCTTGTTGTTGAACTCGACCT. Statistical analysis of the fold change was performed using an Unpaired t-test (GraphPad, San Diego, CA, USA). Each independent experiment was performed with three biological replicates (three pools of 15 mosquitoes) for each condition.

## dsRNA synthesis and injection for gene silencing

Three-to-four day old female *An. stephensi* HP10 females were cold-anesthetized and injected with 69 nl of a three µg/µl dsSOCS, dsHOP, dsDelta, or dsLacZ control. Double-stranded RNA for Suppressors Of Cytokine Signaling (*SOCS*) (ASTE009458 or ASTEI20_041097), Hopscotch (*HOP*) (ASTE008682 or ASTEI20_034867) and Delta (ASTE009642 or ASTEI20_036824) was synthesized by in vitro transcription using the MEGAscript RNAi kit (Ambion, ThermoFisher Scientific, Waltham, MA, USA). DNA templates were obtained by PCR using *An. stephensi* cDNA extracted from whole-body sugar-fed females. A 431-bp fragment was amplified for *SOCS* with primers

containing T7 promoters (F-**TAATACGACTCACTATAGGG** TTCATC-CACTGTCTGGTGCC and R- **TAATACGACTCACTATAGGG** TTTGGTAG CGTCAGCTCGTT), using an annealing temperature of 58 °C. A 498-bp fragment was amplified for *HOP* with primers containing T7 promoters (F-**TAATACGACTCACTATAGGG**CGATGGTGCTAGAATTTCCG and R-**TAATACGACTCACTATAGGG**CGCAGCTCAAACACTCGTAG), using an annealing temperature of 58 °C. A 305-bp fragment was amplified for delta with primers containing T7 promoters (F-**TAATACGAC TCACTATAGGG**GCGGTCAGTCTTGTGAGGAA and R- **TAATACGACTC ACTATAGGG**TTCGTTCTGCTTTCTCGCCT), using an annealing temperature of 58 °C. Double-stranded RNA for lacZ was synthesized by amplifying a 218-bp fragment from lacZ gene clones into a pCRII-TOPO vector using M13 primers to generate a dsRNA control as previously described[28]. Injected mosquitoes were fed in a *P.berghei*-infected mouse to evaluate the effect of gene silencing in the oocyst number. Silencing efficiency with this method ranged between 50 and 60% for all the genes tested in the midgut.

### In vivo live imaging

*An. stephensi* HP10 females infected with *P.berghei* were allowed to lay eggs and then fed another blood meal to distend the midgut. After the second meal, mosquitoes were placed at 19oC again. Mosquito abdomens must be distended to allow visualization of the midgut through the cuticle (transparent lateral panel). We used mosquitoes at two-time points during infection, 10 and 14 days post *P.berghei* infection. Imaging took place the next day at 18–20 h post-bloodmeal. Mosquitoes were imaged as previously described[29]. In short, we removed the legs and heads of 5–10 mosquitoes and placed them between a coverslip and glass slide with craft putty as a spacer. Images were taken on a Leica SP8 confocal microscope using a 40 × 1.25 NA oil objective with a white light laser excitation at 561 nm for td-Tomato and 488 nm for GFP-expressing *P.berghei* oocysts. The z-stack with 1 μm intervals was taken every 10 min for 9–12 h with the resonant scanner. Movies were processed using Imaris 9.9.1 (Bitplane, Concord, MA, USA) and saved as Tiff series. Then, files were imported to ImageJ and converted to the MOV format.

### Viability of oocysts in the midgut

To assess the viability of the oocysts in the midgut, we artificially fed a 10% BSA saline solution described above to engorge the midgut and preserve the morphology for microscopy. Shortly after feeding *P.berghei*-infected midguts from *An. stephensi* HP10 females were dissected and quickly fixed for 30 s in 4% PFA at room temperature. They were then placed in a well containing ice-cold PBS. When a pool of 10 midguts was reached, tissues were transferred to a well containing a LIVE/DEAD™ Fixable Far Red Dead Cell Stain (L34973, ThermoFisher Scientific, Waltham, MA, USA) solution diluted 1:1000 in PBS. Midguts were incubated in the Live/Dead stain solution for 30 min on ice. After that, the tissues were opened longitudinally to form a single sheet, as described above. The bolus content was removed and washed out, and tissues were placed in a 4% PFA fixative solution at room temperature for at least an hour. Finally, midguts were submitted to an immuno-fluorescence protocol to add antibodies to stain either midgut progenitors described above. Live oocysts presented a weaker fluorescent signal in their capsule on the outside, while dead oocysts had a brighter staining inside. Live/Dead fluorescent stain was evaluated on a Leica SP8 confocal microscope using either a 40x or a 63x oil objective, depending on the size of the oocyst with a white light laser with excitation at 633–635 nm for the Live/Dead fluorescent stain. Live/Dead signal is preserved even with detergents present during the immunofluorescence protocol.

### Reporting summary

Further information on research design is available in the Nature Portfolio Reporting Summary linked to this article.

## Data availability

All data that support this study are available from the corresponding authors upon request. The numerical data underlying the plots in the manuscript are provided as a Source Data File. All images and micrographs used in this manuscript are available from the corresponding authors upon request. Source data are provided with this paper.

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

## Acknowledgements

We thank Kevin Lee, Yonas Gebremicale, and André Laughinghouse for insectary support. This work was supported by the Intramural Research Program of the Division of Intramural Research Z01AI000947, NIAID.

## Author contributions

Conceptualization: ABFB, IS, FC, DO, CB-M, Methodology: ABFB, FC, SB, Investigation: ABFB, JS, EB, SB, FC, Visualization: ABFB, SB, Funding acquisition: CB-M, Project administration: CB-M, Supervision: CB-M, IS, DO, Writing—original draft: ABFB, CB-M, Writing—review & editing: ABFB, CB-M.

## Competing interests

The authors declare no competing interests.
