## [Peer Review File · Nature Communications]

Mosquito midgut stem cell cellular defense response limits
Plasmodium parasite infectionREVIEWER COMMENTS

Reviewer #1 (Remarks to the Author):

This manuscript, Mosquito midgut stem cell cellular defense response limits Plasmodium parasite infection, explores a new area in mosquito parasite interactions: epithelial progenitor interaction with oocysts. Using a Gal4 enhancer trap line that lands in the regulatory region of midgut progenitors (and hemocytes), the authors labeled progenitors with the fluorescent reporter, Td tomato, and followed the cells under different conditions including sugar feeding, blood feeding, treatment with the tissue damaging agent bleomycin, RNAi, and Plasmodium infection.

The main observations are 1) an increase in progenitor number and progenitor contact with Plasmodium oocysts at later stages of infection, 5 and 10 days post infection 2) a correlation between treatments that increase and decrease oocyst numbers and the number of progenitors and their association with oocysts 3) direct interactions between reporter expressing cells and oocysts preceding their killing and 4) dead oocysts contained within reporter positive progenitor cells.

Overall, the paper is well-written, the hypothesis is supported by the data, the work represents an important advancement in our understanding of late stage immunity and provides a new mechanism and genetic tool that could be exploited to block Plasmodium infection in mosquitoes.

The authors have addressed the expression in progenitors using a combination of cell morphology and correspondence in expression with the progenitor markers Delta and Klumpfuss. In the supplemental materials, it was shown this HP10 line also traps a hemocyte enhancer. Do the hemocyte dynamics correlate with Plasmodium infection of JAK-STAT pathway manipulation? Do hemocytes also express DI and Klu? Maybe it is unlikely, but is there any possibility that hemocytes can enter the midgut epithelium damaged by Plasmodium or bleomycin?

The observation that progenitors kill and internalize dead oocysts would be further substantiated if the experiments in Figure 4 were repeated under conditions that increase (SOCS kd) or decrease (HOP kd) signaling and progenitor number.

Clarification of the Td tomato reporter. Is this a UAS reporter in the background of the HP10 line? It is also unclear why all the graphs are labeled dsRed volume instead of Td tomato. Same in supplemental methods section "dsRed volume quantification and surface analysis". These are all related, but for consistency, picking one is preferred, except in the methods when talking about specific vendors antibodies.

Fig S3A would be improved with a side view to show more clearly what side of the epithelium they are on, which would be interesting to see and compared to Fig S1.

Provide a reference for this statement: "...expresses a fluorescent reporter (td-Tomato) in a subset of midgut cells with morphology reminiscent of Drosophila midgut stem cells (Fig.1A)..."

For consistency, the microscopy panel in Fig S3C should be labeled progenitor enrichment, not stem cell enrichment

This sentence should be damage (not damaged): We explored whether cell damaged by ookinete invasion was sufficient to elicit proliferation of midgut progenitors

Change GPF to GFP: Dead oocysts and fragments positive for GFP in immunofluorescence

In supplement change mosquitos to mosquitoes: To image the midgut at late stages of infection, we fed mosquitos a saline solution supplemented with 10% BSA (Bovine Serum Albumin) right before dissection to distend the midgut epithelia.

Reviewer #2 (Remarks to the Author):

This concise manuscript by Barletta et al. presents intriguing findings concerning *Plasmodium berghei*-infected *Anopheles stephensi* midguts. The elimination of late-stage oocysts seems to correlate with the degree of stem cell and enterocyte proliferation. While the means of oocyst detection remain unknown, the study demonstrates the influence of the JAK-STAT pathway on this phenomenon, which may be related to earlier findings by the same lab. The experimental procedures are well-documented, controlled and robust, with clear presentation of the results and mostly justified conclusions. Enhancements could be made by addressing the issues noted below.

Major Comments:

The introduction is very brief, encompassing only a single paragraph. It lacks the essential context for comprehending the background and novelty of the outcomes. Many genes and pathways cited for stem cell proliferation require this background to be comprehensible.

The infection assay's setup introduces a crucial variable, temperature, which the manuscript does not sufficiently address. While 28°C is optimal for mosquito physiology and not for *P. berghei* development, the opposite is true for 21°C. This could be tackled by discussing the potential consequences of this discrepancy or by conducting a control experiment to investigate whether the extent and rate of stem cell proliferation is linked to the differing rates of bloodmeal digestion at 21°C (slower and more protracted).

The numbering of panels in Figure 3 is perplexing or possibly erroneous. For instance, A and C (LacZ) and B and D (SOCS) pertain to the same findings. Nevertheless, only A and B are mentioned, with references to C and D pertaining to HOP silencing. This inconsistency is cascaded throughout the panels and should be rectified.

Minor Comments:

In the title, specifying that this phenomenon pertains to a rodent model could enhance the title's clarity. Equally, the discussion must clarify that this is observed in this infection model and that human malaria infection (which occurs at 28°C) may or may not be the same.

In the abstract, insert "is" before "proportional" to correct the grammatical error.

Still in the abstract, it will be beneficial to elucidate how proliferation is experimentally increased or decreased (RNAi silencing of key regulators).

In the 3rd line of the 1st paragraph of main text (introduction), please correct "a motile ookinete" (singular) to "motile ookinetes" (plural) to grammatically match the reference to "female gametes" and "mature oocysts".

Further down the same paragraph, instead of "preventing elimination" describe the capsule's role as protecting from the immune response. You may also further elaborate on the relevance of the complement system in this context.

At the end of introduction, you may replace "dramatic" with "drastic" or "significant" for accuracy, a change that should be mirrored in the results and discussion section.

In Figure 1, utilizing a different color for delta and actin (e.g., not green for both) would prevent confusion.

Consider integrating the data from Figure S3 into the main text, perhaps as additional panels for Figure 1 or as a separate figure, to enhance data comprehension.

The discussion should incorporate references that substantiate the statements that midgut progenitors can move and that oocyst fragments are frequently found within enterocytes.

Reviewer #3 (Remarks to the Author):

The manuscript by Barletta et al. focuses on investigating the interaction between Plasmodium parasites and mosquito midgut progenitors during infection. The authors utilized a transgenic Anopheles stephensi line (HP10) to study the process. Their findings demonstrate that the development of oocysts stimulates the proliferation of midgut progenitor cells. The study further reveals that enhancing the proliferation of progenitor cells can lead to a reduction in oocyst numbers, whereas inhibiting their proliferation promotes the survival of oocysts.

Comments:

The mechanism of how progenitors initially detect parasites is hypothesized but not experimentally tested. More evidence is needed to support hypotheses about the signaling pathways involved.

It is concluded progenitors eliminate oocysts but the distinction between cytotoxic vs. phagocytic functions is unclear from current data. Further tests could provide a resolution.

There is no discussion of specificity, do progenitors respond similarly to non-replicating/non-infectious parasite forms/particles?

The article is written with very little precision and the components are not well organized. There are several errors in the text and figures.

Since the manuscript lacks line numbers, referencing errors or comments accurately becomes challenging.

Main text:

On page 4, "A. stephensi" should be corrected to "An. stephensi" to match the consistent usage throughout the entire manuscript.

Images A-L in Figure 3 are incorrectly referenced in the text compared to the corresponding Figure. This discrepancy should be corrected to ensure accurate and consistent referencing between the text and Figure.

In Figure 4, the legend for image H is missing and should be added for clarity and completeness. Supplementary:

The materials and methods section is poorly written and difficult to understand. It could be reviewed by a language editor to ensure that it is written in clear and concise English, making it more accessible to readers.

The formatting of abbreviations, species and gene names, and other elements should be consistent throughout the entire manuscript and requires a thorough review.

The reference "O'Brochta et al. (2012)" should be cited as a numerical entry, and the remaining citation numbers need to be rearranged.

In general, it is recommended to use "post-infection (PI)" in the figure legends for those evaluated after infection, and "post-feeding" for those that were not infected.

Figure S5 presents the images of blood-fed uninfected midguts. It seems there may be an error in labelling them as "PI" (post-infection). Additionally, there appears to be a repetition of image number (A) in the legend.

REVIEWER COMMENTS

Reviewer #1 (Remarks to the Author):

This manuscript, Mosquito midgut stem cell cellular defense response limits Plasmodium parasite infection, explores a new area in mosquito parasite interactions: epithelial progenitor interaction with oocysts. Using a Gal4 enhancer trap line that lands in the regulatory region of midgut progenitors (and hemocytes), the authors labeled progenitors with the fluorescent reporter, Td tomato, and followed the cells under different conditions including sugar feeding, blood feeding, treatment with the tissue damaging agent bleomycin, RNAi, and Plasmodium infection.

The main observations are 1) an increase in progenitor number and progenitor contact with Plasmodium oocysts at later stages of infection, 5 and 10 days post infection 2) a correlation between treatments that increase and decrease oocyst numbers and the number of progenitors and their association with oocysts 3) direct interactions between reporter expressing cells and oocysts preceding their killing and 4) dead oocysts contained within reporter positive progenitor cells.

Overall, the paper is well-written, the hypothesis is supported by the data, the work represents an important advancement in our understanding of late stage immunity and provides a new mechanism and genetic tool that could be exploited to block *Plasmodium* infection in mosquitoes.

The authors have addressed the expression in progenitors using a combination of cell morphology and correspondence in expression with the progenitor markers Delta and Klumpfuss. In the supplemental materials, it was shown this HP10 line also traps a hemocyte enhancer. Do the hemocyte dynamics correlate with Plasmodium infection of JAK-STAT pathway manipulation? Do hemocytes also express Dl and Klu? Maybe it is unlikely, but is there any possibility that hemocytes can enter the midgut epithelium damaged by Plasmodium or bleomycin?

The reviewer brings up an interesting point. We have never observed hemocytes crossing the basal lamina and coming in direct contact with oocysts in IFAs or live videos. We often observe hemocytes patrolling the basal surface of the midgut, but they do not traverse, even when there are oocysts in close proximity. As suggested by the reviewer, we compare the relative expression of Delta and Klumpfuss mRNAs in hemocytes and progenitor cell-enriched midgut fractions (see below). We found that, in sugar-fed mosquitoes, Delta expression is more than tenfold higher and expression Klumpfuss (Klu) is fourfold higher in midgut progenitors than in hemocytes. We have no evidence of a direct role of hemocytes in late phase immunity against oocysts. However, one cannot rule out that they could release cytokines or other soluble factors that could enhance the response of midgut progenitors to the presence of oocysts. In IFAs, we only observe delta protein expression in midgut stem cells, not in enteroblasts or hemocytes.

The observation that progenitors kill and internalize dead oocysts would be further substantiated if the experiments in Figure 4 were repeated under conditions that increase (SOCS kd) or decrease (HOP kd) signaling and progenitor number.

Point well taken. Although it would be very illustrative, we addressed this question on figure 3, where we manipulate a gene that is specific from stem cells (Delta), to artificially force proliferation and differentiation toward enteroblasts. We observed a decrease in the number of oocysts, which indicates a direct connection between differentiation and proliferation of stem cells and oocyst elimination. Our video images indicate that oocyst killing is a two-step process in which oocysts are first killed by extension of pseudopod-like extensions that come in direct contact with the surface of the oocyst and lyse it, as we observe the dsTomato-positive oocyst cytoplasm being lost towards the midgut lumen. The IFA suggests that this is followed by phagocytosis of parasite remnants by enteroblasts.

Clarification of the Td tomato reporter. Is this a UAS reporter in the background of the HP10 line? It is also unclear why all the graphs are labeled dsRed volume instead of Td tomato. Same in supplemental methods section “dsRed volume quantification and surface analysis”. These are all related, but for consistency, picking one is preferred, except in the methods when talking about specific vendors antibodies.

The HP10 line is an *Anopheles stephensi* transgenic line generated in an SDA-500 wild-type background. This line was generated by the crossing of two transgenic lines on the same background, one containing the enhanced trap Gal4 and another expressing UAS-Td-tomato.

HP10 line was generated using an enhancer trap system described by O'Brochta et al. 2012 (#20 in the reference list). Briefly, *An.stephensi* SDA-500 embryos were injected with a *piggyBac* transposon-based promoter less Gal4 enhancer trap element that was remobilized with *piggyBac* transposase expressed in trans to the gene cassette. That results in random remobilization of the Gal4 containing element. We established remobilized enhancer trap lines that displayed GAL4 expression specifically in adult hemocytes and midgut stem cells. Tissue-specific expression of Gal4 can be visualized by indirect immunofluorescence using anti-GAL4 antibodies or by crossing HP10 to lines containing reporter genes under the regulatory control of GAL4 responsive promoters. In this manuscript we used a crossing of HP10 line to lines that contained a reporter gene (UAS-Td-tomato) under the regulatory control of GAL4.

We have clarified the establishment of the HP10 line in the methods of the paper (lines 207-216 in the revised manuscript).

As suggested, we replaced the term “dsRed volume” with “Td-Tomato” volume in the text. An antibody that recognizes many different red fluorescent proteins (dsRed antibody from Santa Cruz Biotechnology) was used to detect Td-tomato protein expression in IFAs.

Fig S3A would be improved with a side view to show more clearly what side of the epithelium they are on, which would be interesting to see and compared to Fig S1.

Side panels have been added to the Control and Bleomycin treated midguts in figure S3A as suggested by the reviewer.

Provide a reference for this statement: “...expresses a fluorescent reporter (td-Tomato) in a subset of midgut cells with morphology reminiscent of *Drosophila* midgut stem cells (Fig.1A)...”

We have added a reference to the statement (Line 80). Reference number 10.

For consistency, the microscopy panel in Fig S3C should be labeled progenitor enrichment, not stem cell enrichment

We have replaced the term stem cell enrichment for progenitor enrichment in the figure S3C.

This sentence should be damage (not damaged): We explored whether cell damaged by ookinete invasion was sufficient to elicit proliferation of midgut progenitors

The typo has corrected (Line 118 in the revised text).

Change GPF to GFP: Dead oocysts and fragments positive for GFP in immunofluorescence

We have corrected the sentence (Line 165 in the revised text).

In supplement change mosquitos to mosquitoes: To image the midgut at late stages of infection, we fed mosquitos a saline solution supplemented with 10% BSA (Bovine Serum Albumin) right before dissection to distend the midgut epithelia.

We have corrected the sentence (Line 229 in the revised manuscript).

Reviewer #2 (Remarks to the Author):

This concise manuscript by Barletta et al. presents intriguing findings concerning *Plasmodium berghei*-infected *Anopheles stephensi* midguts. The elimination of late-stage oocysts seems to correlate with the degree of stem cell and enterocyte proliferation. While the means of oocyst detection remain unknown, the study demonstrates the influence of the JAK-STAT pathway on this phenomenon, which may be related to earlier findings by the same lab. The experimental procedures are well-documented, controlled and robust, with clear presentation of the results and mostly justified conclusions. Enhancements could be made by addressing the issues noted below.

Major Comments:

The introduction is very brief, encompassing only a single paragraph. It lacks the essential context for comprehending the background and novelty of the outcomes. Many genes and pathways cited for stem cell proliferation require this background to be comprehensible.

Point well taken. We extended the background on the biology of midgut stem cells as suggested by the reviewer (lines 54-73 in the revised manuscript).

“Oocysts develop under the basal lamina of the mosquito midgut epithelium and their surface is covered by a capsule composed of proteins derived from mosquitoes and the parasite³ that is thought to conceal them from the mosquito immune system. Indeed, there is clear evidence that the mosquito complement-like system does not target the oocyst stage⁴. The oocyst is thought to be a “quiet” developmental stage in which the parasite multiples continuously within the capsule. The architecture of the midgut epithelium is intricate in dipteran insects, consisting of at least four distinct cell types: stem cells, which are pluripotent and exhibit high levels of delta protein⁵;

enteroblasts, the partially differentiated cells, that no longer express delta protein but have yet to reach full differentiation⁶. Enteroblasts then give rise to fully differentiated enterocytes and enteroendocrine cells⁶. Enterocytes, the most abundant type, have microvilli and serve both digestive and absorptive functions, while enteroendocrine cells, are involved in hormone-secretion.

Midgut stem cells maintain tissue integrity and homeostasis by self-renewing through asymmetric division. In this process, one daughter cell retains stem cell properties while the other—known as an enteroblast—becomes committed to differentiation^{7,8}. Upon injury, these intestinal stem cells undergo further asymmetric divisions until tissue homeostasis is restored. Most injuries are transient and resolve themselves, leading to a return to the midgut's baseline state^{7,8}. In *Drosophila*, the Jak/STAT pathway plays a pivotal role in controlling midgut stem cell behavior. Its activation triggers both the proliferation of existing stem cells and the differentiation of enteroblasts within the tissue⁹.”

The infection assay's setup introduces a crucial variable, temperature, which the manuscript does not sufficiently address. While 28°C is optimal for mosquito physiology and not for *P. berghei* development, the opposite is true for 21°C. This could be tackled by discussing the potential consequences of this discrepancy or by conducting a control experiment to investigate whether the extent and rate of stem cell proliferation is linked to the differing rates of bloodmeal digestion at 21°C (slower and more protracted).

Point well taken. We had a similar concern, so we first tested whether we observed a strong correlation when we analyzed separately the data from 24 infected mosquitoes from two experiments that were kept at 28°C, and found a strong correlation [$r^2=0.8534$ ($p<0.0001$)] (see graph below). We then analyzed the data from 31 infected mosquitoes that were kept at 21°C and also found a strong correlation [$r^2=0.8018$ ($p<0.0001$)]. When then merged the data from a side by side experiment of mosquitoes fed on the same mouse and that were kept at 21°C or 28°C (total of 43 mosquitoes), and confirmed a strong correlation $r^2=0.9019$ ($p<0.0001$). These are the data shown in Figure 2L. Thus, we conclude that there is a strong correlation between stem cell proliferation and oocysts numbers when mosquitoes are kept either at 21°C or at 28°C. To emphasize this point, the separate analysis of mosquitoes kept at 21°C and at 28°C has been added as an additional supplementary figure (Fig. S6).

The numbering of panels in Figure 3 is perplexing or possibly erroneous. For instance, A and C (LacZ) and B and D (SOCS) pertain to the same findings. Nevertheless, only A and B are mentioned, with references to C and D pertaining to HOP silencing. This inconsistency is cascaded throughout the panels and should be rectified.

Thank you for pointing out this mistake. The text has been rectified to refer the correct panels in Figure 3.

Minor Comments:

In the title, specifying that this phenomenon pertains to a rodent model could enhance the title's clarity. Equally, the discussion must clarify that this is observed in this infection model and that human malaria infection (which occurs at 28C) may or may not be the same.

Thank you for the suggestion. We gave a broad title using “mosquito” to refer to the vector and “*Plasmodium*” to refer to the parasite to reach a broad audience, but in the first line of the abstract we specify that the experiments were done in *Anopheles stephensi* infected with *Plasmodium berghei*. We believe that with this information the reader will know right away which experimental system was used. Although we found that stem proliferation is also proportional to the number of *P. berghei* ookinetes present at 28°C, indicating that this is not an abnormal response of mosquitoes kept at 21°C, we agree with the reviewer that we do not know whether the mosquito midgut progenitors would respond the same to the presence of *Plasmodium falciparum* or *Plasmodium vivax* parasites. To emphasize this point raised by the reviewer, we added the following sentence to the discussion: “It also remains to be established whether these are universal responses of midgut progenitors that are also triggered by infection with human *Plasmodium* parasites.” (lines 188-190 of the revised manuscript)

In the abstract, insert "is" before “proportional” to correct the grammatical error.

Text has been corrected (line 32).

Still in the abstract, it will be beneficial to elucidate how proliferation is experimentally increased or decreased (RNAi silencing of key regulators).

We have changed the sentence according to the suggestion (lines 34-37 in the revised manuscript).

“Silencing components of key signaling pathways through RNA interference (RNAi) that enhance proliferation of progenitor cells significantly decreased oocyst numbers, while limiting proliferation of progenitors increased oocyst survival.”

In the 3rd line of the 1st paragraph of main text (introduction), please correct “a motile ookinete” (singular) to “motile ookinetes” (plural) to grammatically match the reference to “female gametes” and “mature oocysts”.

We have corrected the typos.

Further down the same paragraph, instead of "preventing elimination" describe the capsule's role as protecting from the immune response. You may also further elaborate on the relevance of the complement system in this context.

This information has been added in the introduction (lines 54-57). It now reads: Oocysts develop under the basal lamina of the mosquito midgut epithelium and their surface is covered by a capsule composed of proteins derived from mosquitoes and the parasite³ that is thought to conceal them from the mosquito immune system. Indeed, there is clear evidence that the mosquito complement-like system does not target the oocyst stage⁴.

At the end of introduction, you may replace "dramatic" with "drastic" or "significant" for accuracy, a change that should be mirrored in the results and discussion section.

We have changed dramatic to drastic as suggested (lines 76 and 115).

In Figure 1, utilizing a different color for delta and actin (e.g., not green for both) would prevent confusion.

While we appreciate the reviewer's point, we consistently used red to indicate progenitor cells, but had to use the green channel, the color that gives the best contrast with the black background in IFAs to indicate key elements such as actin (Fig. 1A &B), delta (Fig. 1C&D) or the parasite (Fig. E, f & G). We have clearly indicated what the different staining represent. The combination of colors we used highlights the small triangular cells situated on the basal side of the midgut expressing Delta because they are positive for both red and green staining, and when these colors merged the cells are shown in yellow.

Consider integrating the data from Figure S3 into the main text, perhaps as additional panels for Figure 1 or as a separate figure, to enhance data comprehension.

We appreciate the reviewers point, but Figure 1 is already very complex with panels from A to M and adding many new panels with different types of experiments would be too large and a bit overwhelming. For this reason, we provide all the different lines of experimental evidence that the fluorescent cells are midgut progenitors in one independent supplementary figure (S3).

The discussion should incorporate references that substantiate the statements that midgut progenitors can move and that oocyst fragments are frequently found within enterocytes.

Point well taken. We have incorporated some references describing stem cell migration under the basal lamina of *Drosophila* midguts.

It reads: "Stem cell migration plays a critical role in the effective regeneration of the adult *Drosophila* midgut. Enteroendocrine cells guide the direction of stem cell movement, while enteroblasts support migration by triggering signaling pathways¹⁸. (Lines 190 – 192 in the revised manuscript)

To our knowledge this is the first description of midgut cells containing oocyst fragments, therefore we do not have a reference for that.

Reviewer #3 (Remarks to the Author):

The manuscript by Barletta et al. focuses on investigating the interaction between *Plasmodium* parasites and mosquito midgut progenitors during infection. The authors utilized a transgenic *Anopheles stephensi* line (HP10) to study the process. Their findings demonstrate that the development of oocysts stimulates the proliferation of midgut progenitor cells. The study further reveals that enhancing the proliferation of progenitor cells can lead to a reduction in oocyst numbers, whereas inhibiting their proliferation promotes the survival of oocysts.

Comments:

The mechanism of how progenitors initially detect parasites is hypothesized but not experimentally tested. More evidence is needed to support hypotheses about the signaling pathways involved.

That is a very intriguing question, because at early stages of infection (ookinete) we do not observe a colocalization between parasites and midgut progenitors, which eventually happens at oocyst stage. So, there must be a signal that promotes this encounter. We are currently conducting high throughput analysis, such as bulk and single cell transcriptome to identify potential pathways involved in the detection of the parasites by midgut progenitors. We hypothesize in the text that GPIs could play a role in this signaling but we don't have any evidence of that yet. We are planning to investigate this aspect in further publications.

It is concluded progenitors eliminate oocysts but the distinction between cytotoxic vs. phagocytic functions is unclear from current data. Further tests could provide a resolution.

We concluded that progenitors could eliminate oocyst based on the silencing of regulators of the Jak-STAT pathway that promote either increase or limit progenitor proliferation. With the increase in midgut progenitors, we observed a decrease in oocyst numbers, and when we limit progenitor proliferation that favors oocyst survival.

There is no discussion of specificity, do progenitors respond similarly to non-replicating/non-infectious parasite forms/particles?

Feeding mosquitoes with sugar with Bleomycin (as illustrated in Figure S3A and B) reveals that chemical exposure increases the proliferation of midgut progenitors within 24 hours of treatment. Most of the damaging agents promote an acute and transient increase in midgut progenitors. However, the midgut typically reverts its basal state once the stimulus subsides. A unique feature of midgut *Plasmodium* infection is that it represents a chronic injury to the tissue. Oocyst grow continuously within the tissue for at least 14-15 days until rupture. So, proliferation of midgut progenitors is extended promoting the displacement of epithelial cells and a drastic reorganization of the midgut epithelium (as illustrated in Figure 2A and B).

The article is written with very little precision and the components are not well organized. There are several errors in the text and figures.

Our manuscript is written in a very compact format, but we conducted a detail revision to describe better some sections and add information that was missing in the first version.

Since the manuscript lacks line numbers, referencing errors or comments accurately becomes challenging.

We apologize for the lack of line numbering for reference in the text. We have conducted a detailed revision to ensure that errors in the text and figures were corrected.

Main text:

On page 4, "A. stephensi" should be corrected to "An. stephensi" to match the consistent usage throughout the entire manuscript.

Images A-L in Figure 3 are incorrectly referenced in the text compared to the corresponding Figure. This discrepancy should be corrected to ensure accurate and consistent referencing between the text and Figure.

We have corrected the reference to Figure 3 in the text (lines 147-154).

In Figure 4, the legend for image H is missing and should be added for clarity and completeness.

We have added a description for figure 4H in the legend. Now it reads: "(H) Live oocysts with cyan staining on the surface and a smaller dead oocyst with staining inside of the capsule engulfed by a midgut progenitor. White arrows indicate dead oocysts. Scale Bar: 15 μm ." (Line 659-661).

Supplementary:

The materials and methods section is poorly written and difficult to understand. It could be reviewed by a language editor to ensure that it is written in clear and concise English, making it more accessible to readers.

Point well taken. We have done a detailed revision of the methods of the paper to improve language and readability of the text.

The formatting of abbreviations, species and gene names, and other elements should be consistent throughout the entire manuscript and requires a thorough review.

Point well taken. We have conducted a detailed revision to fix the abbreviations and gene names.

The reference "O'Brochta et al. (2012)" should be cited as a numerical entry, and the remaining citation numbers need to be rearranged.

Thank you for the correction. We have added the numbered reference after the sentence (Line 207). Reference number 20.

In general, it is recommended to use "post-infection (PI)" in the figure legends for those evaluated after infection, and "post-feeding" for those that were not infected.

Point well taken. We have changed the main and supplementary text accordingly.

Figure S5 presents the images of blood-fed uninfected midguts. It seems there may be an error in labelling them as "PI" (post-infection). Additionally, there appears to be a repetition of image number (A) in the legend.

We have corrected the figure and removed the extra text in the legend.

REVIEWERS' COMMENTS

Reviewer #2 (Remarks to the Author):

Thank you for the thorough and thoughtful responses to the comments and revisions to the manuscript. These revisions have significantly improved the manuscript, and I have no further comments.